# Prevalence of Dry Eye Symptoms and Associated Risk Factors among University Students in Poland

**DOI:** 10.3390/ijerph20021313

**Published:** 2023-01-11

**Authors:** Dominika Wróbel-Dudzińska, Natalia Osial, Piotr Witold Stępień, Adrianna Gorecka, Tomasz Żarnowski

**Affiliations:** 1Department of Diagnostics and Microsurgery of Glaucoma, Medical University of Lublin, Chmielna 1, 20-079 Lublin, Poland; 2Faculty of Medicine, Medical University of Lublin, 20-059 Lublin, Poland

**Keywords:** dry eye symptoms, dry eye syndrome, ocular surface, tear film, OSDI, DEQ-5

## Abstract

Aim: The aim of the study was to demonstrate the prevalence and risk factors of dry eye symptoms (DES) among university students in Poland. Material and methods: A cross-sectional study survey was conducted among 312 Polish university students. The questionnaire consisted of the Ocular Surface Disease Index (OSDI), the 5-Item Dry Eye Questionnaire (DEQ-5) and questions regarding medical history and risk factors. Results: According to the OSDI, more than half of respondents (57.1%) have symptoms of ocular surface disease. Time spent using electronic devices is correlated with scores gathered in both OSDI and DEQ-5 (*p* < 0.001). There is a statistically significant dependence between psychotropics (*p* = 0.002), glucocorticosteroids usage (*p* = 0.026), the presence of depression (*p* < 0.001), diabetes (*p* = 0.01) or allergy (*p* = 0.008) and dry eye symptoms proved in both questionnaires. Respondents with refractive errors and those living in metropolitan areas have a statistically higher symptom intensity(*p* < 0.022). Stress felt by students is associated with higher DES risk. No correlation between DES and smoking habits was observed. The history of SARS-CoV-2 infection was associated with the severity of DES (*p* = 0.036). Conclusion: Pathogenesis of DES is multifactorial and its severity depends on several factors, both genetic and environmental. Its prevalence among the young population is underestimated. Determining risk factors will enable the implementation of appropriate prophylaxis and early diagnosis.

## 1. Introduction

Dry eye disease (DED) is one of the most prevalent eye conditions, affecting millions of people globally. The worldwide prevalence ranges from 5% to 50%, depending on the geographic region [1]. DED is defined as a multifactorial eye disease related to loss of the tear film homeostasis. Instability and hyperosmolarity of the tear film leads to damage to the ocular surface, which is associated with ocular symptoms [2]. DED may cause a wide range of ocular symptoms including redness, dryness, discomfort, itching, stinging, burning, irritation, pain, photophobia and foreign body sensation [2,3]. This is typically accompanied by fluctuating vision with blurred or double vision [4,5]. These symptoms can significantly impact patients’ quality of life and lead to reduced work and learning efficiency [4,6]. Patients with DED also have a greater risk of suffering from mental health disorders such as depression and anxiety [7,8]. Despite this, many people with DED remain unevaluated, undiagnosed, and untreated, especially among the young population [9,10].

Previous research has identified many risk factors for DED, including aging, female sex, Asian race, environmental exposures, autoimmune diseases, allergies, hormonal imbalance, psychiatric disorders, certain classes of medications, contact lens wear, refractive surgery, and so on [2,3,10]. However, the extended use of digital display terminals (DDTs), such as computers, smartphones, or tablets, is considered to be a key risk factor for DED, mainly through impaired blinking patterns [11].

Along with the widespread popularization of the use of DDTs for both entertainment and work, and the gradual development of online classes, e-books, or electronic scientific records, the frequency of exposure to DDTs among university students has also increased proportionally. Additionally, the COVID-19 pandemic led students to shift to remote or hybrid learning, further expanding the amount of time they spend using DDTs [12]. Therefore, DED among university students should not be overlooked. Most epidemiological studies on DED have included relatively older populations, over 50 years of age [2]. Data on the characteristics of DED in the youth population, including university students, are limited. Accordingly, it is crucial to carry out research on DED among young people, with particular emphasis on university students.

This study aims to investigate the prevalence and associated risk factors of DES among university students in Poland. Identifying prevalence, symptoms, and risk factors could enable the implementation of appropriate preventive measures against DED as well as the improvement of the diagnostic and therapeutic processes.

## 2. Materials and Methods

Before preparing this article, the literature on the characteristics of dry eye syndrome in a young population, especially students was reviewed. The Medline database was used to search for the following keywords: “dry eye”, “dry eye disease”, “dry eye syndrome”, “dry eye symptoms” and “students”, “college students” and “university students”. There were approximately 30 cohort studies found that have been published since 2012, but they differ in many aspects–including questionnaire types, questionnaire cut-off points and analysis methods. It is worth mentioning that most of these studies came from the pre-COVID era, so their conditions were different from this study. This makes the comparison difficult. However, when we managed to compare the obtained results with the results of previous research, it was pointed out in the discussion section. In summary, based on the search from the present time and current condition, there are not many publications devoted to this topic, hence the idea of this work.

### 2.1. Simple Size Calculation

It is estimated that the total number of university students in Poland in 2022 was 1.2 million, including 40 thousand students in faculties of medicine. The mentioned data come from the report of the Central Statistical Office in Poland. (Central Statistical Office is a central government administration office dealing with the collection and sharing of statistical information on most areas of public life and some aspects of private life.) According to relevant data, the estimated prevalence of DED in the adult population in Poland was 10–18% and up to 30% in certain groups, i.e., mainly in students. Hence, the sampling frame was 30% out of 40,000, i.e., 12,000 individuals. Assuming a margin of error at 5% and a 95% confidence level, the required sample size of the present survey accounted for at least 321 subjects. Despite the data collection process taking a relatively long time, and the effort required, the investigators managed to obtain complete questionnaires from 312 respondents, which constituted 97.2% of the required sample size we previously had computed. The difference did not exceed the 5% threshold, which is why the authors decided to conduct statistical analyses of the answers from 312 respondents.

### 2.2. Study Design

This study was a cross-sectional survey conducted in Poland among university students with no restrictions on study level or field of study. The survey was performed in the form of an available online questionnaire using Google Forms software. Study participants were recruited using a convenience sampling procedure. The URL to the questionnaire was distributed through Facebook and Messenger. Furthermore, snowball sampling was used by encouraging participants to redistribute the survey. By completing the questionnaire participants gave their informed consent to participate in the study. A total of 312 responses were collected. In the form description section of the survey inclusion criteria, the aims and goals of the study were explained. The survey consisted of four sections dedicated to collecting socio-demographic data (gender, age group, place of residence, field of study and year of study), behavioral and environmental factors (use of artificial tears, lifestyle with special emphasis on using electronic devices, spending time outdoors and in air-conditioned interiors), risk factors of dry eye syndrome (refractive error, previous refractive surgery procedures, use and tolerance of contact lenses, history of eye injuries, presence of eye diseases, presence of systemic diseases and taken medications, addictions, history of SARS-CoV-2 infection, vaccination against COVID-19 and stress level) and previously developed and validated dry eye symptom questionnaire tools: the Ocular Surface Disease Index (OSDI) [13] and the 5-Item Dry Eye Questionnaire (DEQ-5) [14] (see Appendix A). The questionnaires use closed single-choice questions. The question about stress level was a self-assessment with scores between 0 to 5, where 0 meant lack of stress and 5 the maximum level of stress.

### 2.3. Dry Eye Symptom Questionnaire Assessment

The Ocular Surface Disease Index (OSDI) OSDI is a 12-item questionnaire that assesses dry eye symptoms occurring during the last week. The questions are divided into three sections which examine, respectively: subjective ocular symptoms, limitations in performing daily activities due to eye problems, and the influence of environmental conditions on eye comfort. Responses to every question ranged from 0, which represents “none of the time”, to 4, which represents “all of the time”. The final score ranges from 0 to 100 (sum of the 12 questions multiplied by 25 and divided by 12, and then rounded to the next integer) with higher scores representing greater disability: normal (0–12), mild (13–22), moderate (23–32), severe (33–100). This study adopted the criteria followed by other authors, whereby symptomatic DED is defined as any OSDI score above 12 [13,15].

The 5-Item Dry Eye Questionnaire (DEQ-5) is a short subset of the full Dry Eye Questionnaire (DEQ) items which contains five questions regarding the frequency and intensity of eye discomfort, eye dryness and watery eyes in the past month. Responses are collected with two types of scales of the answers ranging from 0 to 4 or 5, with 0 corresponding to “never”, and 4 with “constantly” in questions regarding the frequency of symptoms, or 0 with “never have it” and 5 with “very intense” in questions regarding the intensity of symptoms. The final score ranges from 0 to 22, a score > 6 suggests dry eye [14].

### 2.4. Statistical Analysis

Categorical variables were depicted using integer numbers and percentages. Questionnaire scorings were described by their mean, standard deviation, median and lower-to-upper quartile values. The normality of distribution was assessed using the Anderson–Darling test. Statistical significance of differences in the inventory scoring was tested by using generalized linear models due to their non-normal and left-skewed distribution. A multiple regression model was performed in order to estimate the relationship between the OSDI scoring and the DEQ-5 scoring versus a set of independent variables. A multifactor ordinal logistic regression model was carried out for the severity of dry eye according to the OSDI. Spearman’s rank correlation coefficient was also computed for the investigated scorings.

A level of *p* < 0.05 was deemed statistically significant. All the statistical procedures were performed using Statistica^TM^, release 14 (TIBCO Software Inc., Palo Alto, CA, USA).

## 3. Results

The socio-demographic and clinical characteristics of the study cohort are shown in Table 1.

Among the medicaments taken chronically, 15.1% (*n* = 47) respondents use contraceptives, 12.2% (*n* = 38)—psychotropics, 10.6% (*n* = 33)—anti-histamines, 5.8% (*n* = 18)—analgesics, 1.9% (*n* = 6)—beta-blockers, 1.6% (*n* = 5)—glucocorticosteroids, 1.0% (*n* = 3)—anabolic steroids and 10.9% (*n* = 34)—other hormones. Among comorbidities, the most frequently reported one was allergy (24.7%, *n* = 77), then ex aequo thyroid disease and depression (11.9%, *n* = 37), followed by hormonal disorders (8.3%, *n* = 26), mental diseases (5.4%, *n* = 17), autoimmune disease (4.2%, *n* = 13), acne (4.2%, *n* = 13), arterial hypertension (3.2%, *n* = 10) and diabetes (2.6%, *n* = 8). Medication taken chronically and comorbidities are shown in Table 2 and Table 3, respectively.

In a 0–5 scale, where the bigger the number, the higher the stress level, majority of respondents (36.9%, *n* = 115) assessed their stress level as 3, then 27.9% (*n* = 87) as 4, 20.8% (*n* = 65) as 2, 7.7% (*n* = 24) as 1, 6.1% (*n* = 19) as 5. Only two people (0.6%) chose number 0. (Figure 1). Mean value of the results: 3.0, median value: 3.0, standard deviation: 1.0, 95% CI: 2.9–3.1.

### 3.1. Dry Eye Symptoms Prevalence

The mean OSDI value was 20.3 ± 17.4 (95% CI: 18.3–22.2) and the mean DEQ-5 value was 7.5 ± 4.8 (95% CI: 6.9–8.0). Both of these values are greater than the defined cut-off points (OSDI scores > 12 and DEQ-5 scores > 6) for diagnosis of DED. Based on the OSDI questionnaire, the prevalence of DES among the study population was 57.1%. According to the OSDI score grading, less than half of respondents were classified with normal ocular surface (42.9%, *n* = 134), 24.7% (*n* = 77) with mild, 14.4% (*n* = 45) with moderate and 18.0% (*n* = 56) with severe ocular surface disease.

### 3.2. Socio-Demographic and Clinical Risk Factors

Dry eye syndrome is more common in women, which was also observed in our study. The female sex is associated with a higher score than the male sex, meaning more severe dry eye symptoms. Though the mean score for both groups is classified as mild intensity (F: 21.9; M: 16.5), the median score for males (10.4) corresponds to normal eye comfort (F: 16.7). Residency in metropolitan areas corresponds to higher symptom intensity than in rural areas (*p* = 0.022; mean 21.3 vs. 16.8; median 16.7 vs. 12.7, respectively). No refractive error is connected with statistically lower OSDI result (*p* < 0.001). Good contact lens tolerance is related to a lower OSDI score, unlike contact lens intolerance (*p* < 0.001; mean 19.0 vs. 36.7; median 16.7 vs. 37.5, respectively). No statistical correlation was found between the OSDI score and the field of study, history of SARS-CoV-2 infection and vaccination against COVID-19.

There is a correlation between gender and DEQ-5 as well (*p* < 0.001). Females have higher scores than men (mean: 8.1 vs. 6.0, median: 8 vs. 5). Factors associated with lower symptom intensity according to the DEQ-5 are no refractive error (*p* < 0.001) or myopia alone (*p* = 0.042). Contact lenses correlate with more severe dry eye symptoms (*p* = 0.028), though users with good contact lens tolerance have less severe symptoms than people who do not tolerate them (*p* < 0.044; mean: 7.6 vs. 10.1; median: 8 vs. 11). Differences between the metropolitan area and rural residents were close to the statistical difference (*p* = 0.052) with higher scores in the former. Comparably to the OSDI questionnaire analysis, there was no correlation between the DEQ-5 score and the field of study, history of SARS-CoV-2 infection and vaccination against COVID-19.

### 3.3. Behavioural and Environmental Risk Factors

Time spent using electronics is correlated with scores gathered in both OSDI and DEQ-5 (*p* < 0.001). In OSDI at least 4 h of exposure causes mild eye discomfort (mean: 19.6, median 16.7), while 8 h—moderate discomfort (mean: 24.9; median: 18.8). A four-hour or longer exposure to electronic screens is associated with dry eye symptoms, according to DEQ-5. The longer a device is used without any breaks, the higher scores in both the OSDI (*p* = 0.005) and the DEQ-5 (*p* = 0.058). A reverse trend was observed for the overall time spent outdoors daily, though no strict correlation was found (OSDI: *p* = 0.065, DEQ-5: *p* = 0.057). There was no correlation between DES and time spent in air-conditioned interiors (OSDI: *p* = 0.376; DEQ-5: *p* = 0.568) as well as regular, daily basis use tobacco products (OSDI: *p* = 0.369; DEQ-5: *p* = 0.641).

### 3.4. Medications and Comorbidities

Taking certain drugs and the presence of some comorbidities have an influence on the statistically higher score in the OSDI compared to subjects who do not use those medicaments or do not suffer from those diseases: psychotropics (*p* = 0.002), anti-histamines (*p* = 0.024), glucocorticosteroids (*p* = 0.024) and depression (*p* < 0.001), mental disease (*p* = 0.019), diabetes (*p* = 0.008), allergy (*p* = 0.002). After multifactor analysis, the stress level (*p* = 0.012), refractive error (*p* < 0.001) and mental diseases (*p* = 0.038) were significantly associated with a greater risk of DES.

Similar to the OSDI, results show that certain drugs admission and the presence of some comorbidities have an influence on a higher score in the DEQ-5: psychotropics (*p* = 0.014), glucocorticosteroids (*p* = 0.026), contraceptives (*p* = 0.023) and depression (*p* = 0.005), diabetes (*p* = 0.001), allergy (*p* = 0.023). The effect of taking anti-histamines was close to significance (*p* = 0.054). The comorbidities and medications taken by the respondents are shown in Table 2 and Table 3, respectively.

Moreover, multifactorial analysis reveals that stress level (*p* = 0.009), refractive error (*p* < 0.001), gender (*p* = 0.020) and place of residence (*p* = 0.046) correlated with higher risk of DES.

Severity of dry eye, according to the OSDI, depending on the behavioral and environmental factors, medicaments used and occurrence of comorbidities are shown in Table 4, Table 5 and Table 6.

Multifactor analysis showed that the greatest influence on the higher score both in the OSDI and DEQ-5 was stress level (respectively *p* = 0.012 and *p* = 0.009). Mental diseases were proven to be a significant factor in the OSDI multifactor analysis (*p* = 0.038), while in the DEQ-5 multifactor analysis, other factors showed to be significant were gender (*p* = 0.020) and place of residence (*p* = 0.046).

### 3.5. Correlation of the OSDI and DEQ-5 Questionnaires

There was a significant correlation between the OSDI and DEQ-5 scores, based on Spearman’s rank correlation coefficient (rho = 0.82, *p* < 0.001). Figure 2 presents a scatterplot of OSDI against DEQ-5.

## 4. Discussion

Despite an ever-growing interest in dry eye syndrome and the increase in epidemiological studies assessing DED worldwide, most of them have included relatively older populations, over 50 years of age [2]. Moreover, the vast majority of them came from the pre-COVID era; nowadays, we live in completely different conditions. There is still a significant lack of updated knowledge about DES among young people, especially university students. One of the main risk factors of DES is the long-term use of visual display terminals, such as smartphones, tablets or computers [16], which is very prevalent among children and adolescents [17]. Accordingly, DED among the youth population should not be neglected. This study aimed at evaluating the prevalence of symptoms and identifying risk factors for dry eye among university students in Poland.

### 4.1. Dry Eye Disease Prevalence

Based on the OSDI cut-off point for the diagnosis of DED (>12), the prevalence of DES among the study population was 57.1%. According to the OSDI score grading, 134 (42.9%) participants were asymptomatic, 77 (24.7%) had mild dry eye symptoms, 45 (14.4%) had moderate dry eye symptoms and 56 (18.0%) had severe dry eye symptoms. The results are in line with recent studies carried out in other countries. Cross-sectional surveys, based on the same diagnosis criteria (OSDI score > 12), showed that the prevalence of DED among, Brazilian [18], Serbian [19] and Dubai [20] students was 50.5%, 59.64%, 60.5% and 62.6%, respectively. This resemblance might be due to the similarity of exposure factors between evaluated populations, such as age level, student lifestyle and amount of time spent using DDTs associated with both studying and entertaining.

### 4.2. Socio-Demographical and Clinical Factors

Age and the female sex have been found to be the greatest risk factors for dry eye [2]. Contrary to previous studies [2,21], there was no significant correlation between dry eye symptoms and age, possibly due to slight age differences between study participants. This study involved young people with a narrowed age range; 76.3% of the respondents were aged 20–25, 14.1% were 26–30 years old and only 5.4% and 4.2% were below and above these age groups, respectively.

However, like other previous studies on DED and ocular surface diseases [2,22], there was a higher prevalence and more severe symptoms in women (OSDI: 21.9 ± 17.6; DEQ-5: 8.1 ± 4.1) than in men (OSDI: 16.5 ± 16.4; DEQ-5: 6.0 ± 4.7) (*p* = 0.001); but in this study, the vast majority of respondents were women 70.2%. Females are more vulnerable to DED due to the effects of sex hormones (e.g., androgens, estrogens), hypothalamic-pituitary hormones, glucocorticoids, insulin, insulin-like growth factor 1 and thyroid hormones, as well as to the sex chromosome complement, sex-specific autosomal factors and epigenetics [23]. Women are also more prone to systemic conditions that promote DES, such as autoimmune diseases, allergies, or psychiatric disorders [24]. Recently, Sonkodi B. has also proposed the molecular basis of the sex difference in the epidemiology of DED, suggesting the role of nerve growth factor (NGF)-tropomyosin receptor kinase A (TrkA) axis signaling [25]. Additionally, oral contraceptive use is considered a risk factor for DES [26,27]. In this study, nearly 22% (47 out of 219) of women declared using oral contraceptive pills (OCP).

Regarding factors associated with DES, place of residence plays a significant role. Students living in the cities have shown more symptoms than students living in rural areas (OSDI: *p* = 0.022; DEQ-5: *p* = 0.052). This may be associated with greater exposure to adverse environmental factors, such as air pollution; prolonged stay in confined, poorly ventilated rooms; higher levels of stress and an unhealthy diet containing highly processed foods. Our results correlate with findings in Korean [28] and Indian [29] populations, but are not consistent with the studies in Ghana [30], where the urban population is at lower risk of DED. However, there is a huge disparity in health and eye care service provision and utilization patterns between rural and urban areas in Ghana [31], which may play a substantial role in the different distribution of DED among the Ghanaian population, compared to other more urbanized countries.

In this study, respondents with any or a combination of refractive errors had a greater risk of DES than those without refractive errors (*p* < 0.001). This was consistent with research carried out on Saudi Arabian [32] and Trinidad and Tobago students [33]. Considering that uncorrected refractive errors could lead to ocular discomfort, more frequent DED symptoms among students with these issues may be associated with wrong eyeglasses prescriptions or non-compliance with the recommendation for spectacle correction. Additionally, in this study, nearly 45% (93 out of 207) of students with refractive errors reported using contact lenses, a known risk factor for DES [2], which may have influenced overall results in this group.

As compared to participants who did not report using contact lenses, contact lens wearers were more likely to have DES, consistent with previous studies [2,34,35]. Contact lenses correlated with more severe dry eye symptoms (DEQ-5: *p* = 0.028; OSDI: *p* = 0.056)), though users with good contact lens tolerance had less severe symptoms than people who do not tolerate them (OSDI: *p* < 0.01; DEQ-5: *p* = 0.04). When the contact lens is worn, the tear film becomes separated into the pre- and post-lens tear film, making it unstable and prone to evaporation [35]. The resulting friction between the lens and the eye damages the ocular surface. In addition, decrease in aqueous volume in the pre-lens tear film and mechanical stimulation to the Meibomian glands leads to increased Meibomian gland dropout. Without proper functioning of the Meibomian glands, the tear fluid is deprived of the constituents of the lipid layer, responsible for stabilizing the tear film, which leads to loss of the tear film homeostasis and triggers ocular symptoms [36].

### 4.3. Behavioral and Environmental Factors

Extended use of DDTs is considered to be a key risk factor for DED, which has been confirmed by many studies conducted in various study populations [2,37,38], and also in students in other countries [18,19,20,32,33]. DDT use triggers impaired blinking patterns, leading to disturbed meibum distribution and reduced exposure of the eye surface to tear film, which causes ocular surface damage [11,37]. The results of the present study correspond with previous findings. An increase in the number of hours a day spent on DDTs significantly increases the risk of DES in students (OSDI: *p* = 0.001; DEQ-5: *p* < 0.001). In addition, the less frequent breaks while using the screen also correlate with a higher risk of DES (OSDI: *p* = 0.01: DEQ-5: *p* = 0.058). Moreover, both of these factors significantly contribute to the severity of DES, according to the OSDI grading (screen time: *p* = 0.006; frequency of breaks: *p* = 0.016). These findings are consistent with recent research, assessing various types of tools for controlling the amount of time spent on DDTs and reminding people to take regular breaks. Techniques that encourage taking breaks, such as animations for computer users, have been shown to reduce dry eye symptoms [39,40].

Other environmental and behavioral factors that have previously been suggested to contribute to DED include the amount of time spent outdoors and in air-conditioned interiors during the day [2]. This analysis showed no statistically significant correlation between the DES and these factors among this study population. However, there was a clear, strong trend towards an increase in the prevalence of DES as the time spent outdoors decreased (OSDI: *p* = 0.065; DEQ-5: *p* = 0.057). Similarly, a large-scale population-based study of 40,501 people in the Netherlands found that participants working outdoors were less likely to suffer from dry eyes than those working indoors [41]. This can be explained by an unfavorable indoor environment, which encompasses a combination of air quality, bioaerosols, relative humidity, airflow and temperature [42].

It has also been suggested that a smoking habit is a risk factor for DED [2,43]. Nevertheless, the results of this study did not reveal any association between DED and regular use of tobacco products on a daily basis. Additionally, studies in Serbia [19] and Dubai [20] did not find a smoking habit to be a risk factor for DED among university students. Contrarily, studies on Malaysian [44] students showed a significant association between DED and smoking. These discrepancies may be due to the differences in the length of smoking time, the number of cigarettes smoked per day and the use of various forms of smoking (conventional cigarettes, electronic cigarettes) [45]. Therefore, the effect of smoking on DED remains inconclusive and more research is needed to understand and establish the role of smoking in the onset of DED.

### 4.4. Comorbidities and Medications

Along with the many risk factors for DES, comorbidities and medications taken chronically have been implicated in the onset and persistence of DED [2,10]. In this study population, there was a significant association (*p* < 0.05) between DES and stress level, mental diseases, depression, allergy and diabetes as well as psychotropics, anti-histamines, glucocorticosteroids and oral contraceptives.

DED can be affected by stress [7,46]. A correlation between perceived stress level and dry eye symptoms was found in multifactorial analysis of this study. It was described that stress and psychiatric diseases such as depression or mood disorders can result in more aggravated DED than it would imply from tear film dysfunction and changes observed in ophthalmological examination [46]. The association between DED and stress level may be due to an increase in the production of interleukin-1, -2, -6, -8 and TNF-alpha. These factors lead to both ocular surface inflammation and the intensification of negative emotional status [7].

In this study, it was found that depression and other mental diseases were significantly associated with a greater risk of DES (OSDI: respectively, *p* < 0.001 and *p* = 0.019, DEQ-5: respectively, *p* = 0.005 and *p* = 0.127, which is close to significance), as well as increased severity of DES (OSDI: respectively, *p* < 0.001 and *p* = 0.039). These results are consistent with previous studies on the connection between DED and mental illnesses (including depression, anxiety, PTSD, dementia, bipolar disorder and neurotic disorders) [8,47,48] and also in the population of young adults [49,50]. Moreover, depression and anxiety were found to be more prevalent and their symptoms were significantly more severe in patients diagnosed with DED [51]. However, it is suggested that psychiatric symptoms such as depression or anxiety are related only to subjective symptoms of DED but not objective symptoms [52]. The relationship between mental health and DED symptoms is most likely bidirectional, as both these disorders influence each other. Interestingly, the tears of patients with depression showed higher levels of inflammatory cytokines IL-6, IL-17 and TNF-a, which may take part in dry eye inflammation [53]. It is also worth mentioning that dysregulation of serotonin levels in depression may induce dry eye by disturbances in the ocular surface as serotonin receptors are present in the conjunctival epithelium and Meibomian glands [54].

Another factor connected to mental health that our study showed to be significantly associated with a greater risk of DES (OSDI: *p* = 0.002, DEQ-5: *p* = 0.014) and increased severity of DES (OSDI: *p* = 0.002) is psychotropics usage. Previous studies showed that antidepressant usage is associated with DED independently of depression itself [55]. Serotonin reuptake inhibitors such as SSRI and SNRI, which are used as a first-line treatment for depression, were proven to induce and increase the severity of dry eye. Mechanisms involved in this process include competing with acetylcholine at postsynaptic muscarinic receptors and reducing the signaling of tear component secretion resulting in tear film instability but also raising serotonin levels in the tear film which can contribute to ocular surface inflammatory markers and apoptosis in human corneal epithelial cell culture [56]. As was shown in a recent study, patients using SNRIs have lower OSDI scores than patients using SSRIs or TCAs. It suggests that among the most common antidepressants, SNRIs seem to have the least exacerbating impact on dry eye symptoms, which can be caused by their pain-reducing properties as they are also used in the treatment of chronic pain syndromes [57]. Other psychotropics were also shown to be associated with DED. Lithium, used in the treatment of bipolar disorder is suggested to induce dry eye by decreasing tear production and disrupting the tear film stability [58]. It is also suggested that dry eye symptoms may be related to the usage of anti-psychotics, with a higher risk of DED associated with the usage of first-generation anti-psychotics than second-generation anti-psychotics [48]. These findings indicate the need to pay special attention to the mental health of patients with DED.

Previous studies showed that allergy and use of anti-histamines are associated with the higher risk of DED [47,59]. These findings are consistent with our study in which allergy was significantly correlated with the higher risk (OSDI: *p* = 0.002, DEQ-5: *p* = 0.023) and severity (OSDI: *p* = 0.008) of DES and usage of anti-histamines was significantly correlated with higher risk of DES (OSDI: *p* = 0.024, DEQ-5: 0.054 which is close to significance) while its effect on the severity of DES was close to significance (OSDI: *p* = 0.156). Clinical manifestations of DED and ocular allergy include overlapping signs and symptoms, they can coexist or predispose to each other. Moreover, there is evidence of shared pathogenetic pathways between them [60]. Ocular allergy was shown to be associated with tear film instability by changes of meibomian glands, lipid layer and mucins alterations. It is also suggested to induce tear film hyperosmolarity. Ocular surface inflammation in ocular allergy involves proteolytic enzymes such as MMP-9, which plays an important role in DED pathogenesis and leads to corneal epithelial damage. These abnormalities contribute to neurosensory abnormalities (reduction of corneal sensitivity) which impact tear secretion, blink rate, epithelial and goblet cell tropism and the behavior of the corneal immune cells [61]. Corneal lymphangiogenesis, which was found in DED and ocular allergy, facilitates immune cell activation, enhances pro-inflammatory cytokines and damages corneal nerves [62]. Moreover, both DES and ocular allergies affect conjunctiva-associated lymphoid tissue, which contributes to the ocular surface damage and self-continuation of the inflammation [63]. Increased risk of DES was documented in patients with asthma. This correlation may be connected to usage of antihistamines, adverse environmental conditions affecting both the ocular surface and the airways and possible alteration of ocular homeostasis by inflammatory processes present in asthma [64]. Patients with atopic dermatitis are significantly more likely to experience dry eye, possibly due to genetic susceptibility, trauma from excessive rubbing and therapies for atopic dermatitis [65]. Anti-histamines contribute to DED, as they exhibit antimuscarinic effects on peripheral muscarinic receptors, which leads to decreased tear production by decreasing aqueous output from the lacrimal glands and mucin output from the goblet cells. They also may induce vasoconstriction and alter blood flow to lacrimal glands. These effects are more typical of first-generation anti-histamines, as they are less selective to histamine receptors [66].

As in previous studies [2,26,27], the results obtained herein also showed that women using oral contraceptives are at greater risk of suffering from DES compared to women not using this type of contraception. Despite the widespread clinical perception that oral contraceptives may be associated with DED symptoms, there are surprisingly few studies examining the pathophysiology of this relationship [23,26]. It is suspected that increased estrogen due to the use of oral contraceptives may cause a decreased production of lipid components of the tear film. Researchers have also suggested that the decrease in serum androgen levels that occurs during the use of oral contraceptives may trigger the development of a non-immune type of DED [25].

Diabetes mellitus was previously documented to be associated with the risk of DED through altering pathways of lacrimal gland secretion and protection of the ocular surface [67]. Although this study is consistent with these findings and showed that diabetes is associated with higher risk (OSDI: *p* = 0.024, DEQ-5: *p* = 0.001) and severity of DES (OSDI: *p* = 0.010), it is difficult to decide whether these results are significant in the study cohort because of the small number of respondents who reported the presence of diabetes (*n* = 8).

Correlation of steroid use with DES was presented in previous studies, which was significant with inhaled but not with oral steroid use [68]. In this study, steroid use was associated with higher risk of DES (OSDI: *p* = 0.024, DEQ-5: *p* = 0.026). However, similarly to diabetes, the number of participants who reported taking steroids was small (*n* = 5), making it difficult to determine whether these results are significant in our study cohort.

Interestingly, there was no correlation between DES and autoimmune, thyroid and hormonal diseases, which are widely recognized as risk factors [2,10,47]. DES has been linked to autoimmune diseases, such as rheumatoid arthritis, systemic lupus erythematosus, systemic sclerosis, and especially Sjögren syndrome, both primary and secondary [69,70,71]. The unbalanced regulatory mechanism of protective immunity at the ocular surface may cause conjunctival squamous metaplasia, loss of goblet cells and the formation of mucus aggregates, which in turn can disrupt tear film production and flow [71]. Guannan et al. showed that the cell injury on the ocular surface was more serious in subjects with dry eye in systemic autoimmune disease than in subjects with dry eye in healthy controls [69]. Likewise, thyroid diseases, which in most cases are also autoimmune disorders, have been identified as a risk factor for DED. Dry eye syndrome in patients with Hashimoto’s thyroiditis is believed to result from meibomian gland dysfunction and is correlated with the duration of the thyroid disease [72]. On the other hand, thyroid orbitopathy in the course of Graves’ disease is connected with eyelid retraction, proptosis, lid lag, and/or restrictive extraocular myopathy, which can each contribute to ocular surface symptoms and, in extreme cases, can lead to severe exposure keratopathy and even corneal ulceration [73]. Scientists have identified many hormone-related mechanisms in the pathogenesis of DED and considered hormonal disorders as an important risk factor for DED [23,74]. This difference between the studies can be explained by the younger age of the participants and the resulting shorter duration of the disease. Ocular symptoms in the course of mentioned diseases, usually appear relatively later as secondary lesions to the underlying disorder. Unlike older adults, who often have comorbidities, younger people respond better to treatment and more easily achieve and maintain metabolic control, which prevents the development of DES. Furthermore, nearly half (45.8%) of this study’s participants were medical students, which may be related to greater awareness and better compliance in the treatment of both underlying disease and ocular symptoms.

Intriguingly, the history of SARS-CoV-2 infection was associated with the severity of DES, based on the OSDI grading (*p* = 0.036). Students who recovered from COVID-19 reported more severe symptoms than respondents without a history of the infection. The vast majority of studies evaluating the impact of COVID-19 on DES have focused on the social context of the pandemic, such as lifestyle changes, the shift to remote learning and the associated extended use of electronic devices (computers, tablets, smartphones). Several studies assessing various manifestations of COVID-19 have shown that most patients have ocular symptoms during active infection [75,76,77]. There is a need to evaluate the long-term impact of COVID-19 infection on DES in order to investigate the possible negative effects on the ocular surface.

### 4.5. Correlation of the OSDI and DEQ-5 Questionnaires

There was a significant correlation between the OSDI and DEQ-5 scores, based on Spearman’s rank correlation coefficient (rho = 0.82, *p* < 0.001), which means that the overall results of the OSDI are related to dry-eye symptoms measured by DEQ-5 (*p* < 0.001). It confirms the reliability of the questionnaires used in the study and the credibility of the respondents.

The specific, validated and objective clinical tests for dry eye disease assessment were not performed, which is a limitation of this study. The conducted survey focused only on the reported subjective symptoms of dry eye. Bearing in mind that asthenopia may provoke some dry eye symptoms, and conducting a survey without a proper diagnosis, may result in overlapping in the reports. There are some patients with dry eye syndrome without subjective symptoms. Because the objective clinical tests were not performed, this group of patients may have been omitted from this work, which is another limitation of the study. However, it is worth mentioning that many studies show a lack of independence between subjective and objective symptoms, which is why this work is focused on subjective symptoms affecting the comfort and quality of life of the respondents.

## 5. Conclusions

Our study revealed that DES is a very common disease among young adults, affecting 57.1% of students in Poland. We found that the female sex, urban areas, refractive errors, contact lens use, extended screen time, the lack of breaks when using screens, a certain class of medication (psychotropics, anti-histamines, contraceptives) and some comorbidities (depression, mental diseases, allergies) are the risk factors for DES among students. The results correspond with a growing pool of evidence that DES are more common in young adults, especially in times of the popularization of remote learning and extending the time of using digital display terminals. Accordingly, it is crucial to increase awareness about this condition and continue research on DES among the younger generation to obtain a more detailed analysis. Identifying the prevalence, symptoms and risk factors could enable the implementation of appropriate preventive measures against DED among the young population.

## Figures and Tables

**Figure 1 ijerph-20-01313-f001:**
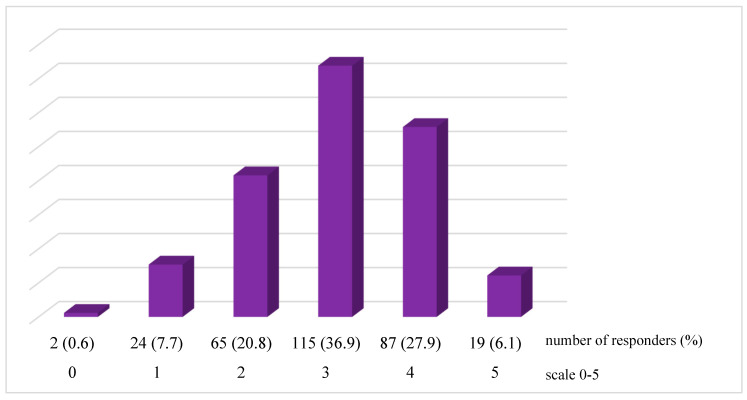
Stress level among the responders (*n* = 312).

**Figure 2 ijerph-20-01313-f002:**
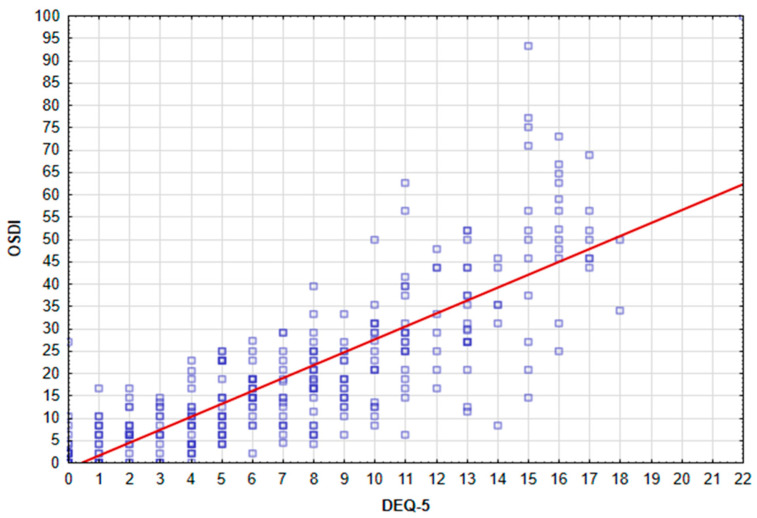
Scatterplot of OSDI against DEQ-5.

**Table 1 ijerph-20-01313-t001:** Baseline socio-demographic and clinical characteristics of the study cohort (*n* = 312).

Investigated Trait	*n*	%
Gender:		
Female	219	70.2
Male	93	29.8
Age group (years):		
<20	17	5.4
20–25	238	76.3
26–30	44	14.1
>30	13	4.2
Place of residence:		
Metropolitan area	241	77.2
Rural area	71	22.8
Field of study (faculty):		
Medicine	143	45.8
Non-medical	169	54.2
Refractive errors:		
None	108	34.9
Myopia (alone)	94	30.4
Hyperopia (alone)	13	4.2
Astigmatism (alone)	24	7.8
Myopia and astigmatism	67	21.7
Hyperopia and astigmatism	3	1.0
Use of contact lenses (overall):	93	29.8
≤1 year	19	20.4
2–3 years	17	18.3
Over 3 years	57	61.3
Contact lens tolerance:		
Good	79	85.9
Not good	13	14.1
Regular, daily basis use tobacco products	74	23.7
History of SARS-CoV-2 infection	173	55.4
Vaccination against COVID-19		
No	41	13.1
Full dose, and a third booster	133	42.6
Full dose (2 of two-dose, or 1 single-dose vaccine)	134	43.0
Incomplete dose (1 dose of two-dose vaccine)	4	1.3

**Table 2 ijerph-20-01313-t002:** Medication taken chronically by the study participants (*n* = 312).

Pharmaceutical Agent	Frequency		DEQ-5 Score			OSDI Score		
	*n*	%	*M*	*SD*	*p*-Value	*M*	*SD*	*p*-Value
Beta-blockers	6	1.9	11.2	6.6	=0.159	31.6	35.4	=0.482
Psychotropics	38	12.2	9.4	5.2	=0.014	30.1	23.1	=0.002
Anti-histamines	33	10.6	9.4	5.7	=0.054	28.5	23.3	=0.024
Analgesics	18	5.8	7.7	5.8	=0.923	30.4	26.7	=0.062
Glucocorticosteroids	5	1.6	13	5.2	=0.026	44.8	33.4	=0.024
Contraceptives	47	15.1	9.1	5.3	=0.023	24.7	20.6	=0.127
Other hormones	34	10.9	8.8	5.4	=0.170	26.7	23.7	=0.152

*M*—mean; *SD*—standard deviation.

**Table 3 ijerph-20-01313-t003:** Comorbidities in the study participants (*n* = 312).

Health Condition	Frequency		DEQ-5 Score			OSDI Score		
	*n*	%	*M*	*SD*	*p*-Value	*M*	*SD*	*p*-Value
Arterial hypertension	10	3.2	10.1	6.2	=0.165	29.6	29.0	=0.299
Heart disease	6	1.9	9.0	7.5	=0.778	30.7	35.7	=0.620
Depression	37	11.9	9.7	5.2	=0.005	30.6	20.8	<0.001
Mental disease	17	5.4	9.6	5.9	=0.127	34.0	28.0	=0.019
Thyroid disease	37	11.9	8.5	5.4	=0.219	26.8	24.8	=0.155
Diabetes	8	2.6	13.9	5.1	=0.001	44.8	29.7	=0.008
Allergy	77	24.7	8.7	4.7	=0.023	25.4	19.6	=0.002
Acne	13	4.2	9.5	6.2	=0.242	30.3	29.0	=0.210

*M*—mean; *SD*—standard deviation.

**Table 4 ijerph-20-01313-t004:** Severity of dry eye, according to the OSDI, in the study participants by selected behavioral and environmental factors (*n* = 312).

Independent Trait	Normal		Mild		Moderate		Severe		*aOR* *p*-Value
	*n*	%	*n*	%	*n*	%	*n*	%	
How many hours a day do you spend in close proximity to the screen of an electronic device, e.g., mobile phone, computer, tablet, reader (not taking into account a TV set)?									
<4	29	21.6	11	14.3	6	13.3	3	5.4	=1.37 =0.006
4–8	65	48.5	49	63.6	24	53.4	25	44.6	
>8	40	29.9	17	22.1	15	33.3	28	50.0	
How many hours a day do you spend in front of the screen of an electronic device (not taking into account a TV set), without taking breaks?									
<1	32	23.9	6	7.8	8	17.8	5	8.9	=1.16 =0.016
1–2	56	41.8	43	55.8	19	42.2	23	41.1	
3–4	26	19.4	23	29.9	11	24.4	19	33.9	
>4	20	14.9	5	6.5	7	5.6	9	16.1	
How many hours a day do you spend outdoors?									
<1	39	29.1	25	32.5	16	35.6	21	37.5	=0.261
1–4	75	56.0	48	62.3	22	49.9	31	55.4	
>4	20	14.9	4	5.2	7	15.6	4	7.1	
How many hours a day do you spend in air-conditioned interiors?									
<1	83	61.9	44	57.1	24	53.3	34	60.8	=0.648
1–4	30	22.4	21	27.3	9	20.0	11	19.6	
>4	21	15.7	12	15.6	12	26.7	11	19.6	
Regular, daily basis use tobacco products	32	23.9	11	14.3	15	33.3	16	28.6	=0.078
History of SARS-CoV-2 infection	75	56.0	34	44.2	25	55.6	39	69.6	=0.036

*aOR*—adjusted odds ratio; the adjutancy procedure comprised the participants’ age, gender, place of residence, faculty, refraction error, use of contact lenses, comorbidities, medicines taken.

**Table 5 ijerph-20-01313-t005:** Severity of dry eye, according to the OSDI, in the study participants by medication (*n* = 312).

Pharmaceutical Agent	Normal		Mild		Moderate		Severe		*aOR* * *p*-Value **
	*n*	%	*n*	%	*n*	%	*n*	%	
Psychotropics	9	6.7	8	10.4	6	13.33	15	26.8	=1.41 =0.002
Anti-histamines	12	9.0	5	6.5	6	13.3	10	17.9	=0.156
Analgesics	6	4.5	3	3.9	4	8.9	5	8.9	=0.433
Contraceptives (females only)	19	23.7	8	12.5	7	20.6	12	29.3	=0.184
Other hormones	13	9.7	7	9.1	5	11.1	9	16.1	=0.572

* *aOR*—as above. ** Subjects who used an agent versus the subjects who did not.

**Table 6 ijerph-20-01313-t006:** Severity of dry eye, according the OSDI, in the study participants by comorbidity (*n* = 312).

Health Condition	Normal		Mild		Moderate		Severe		*aOR* * *p*-Value **
	*n*	%	*n*	%	*n*	%	*n*	%	
Arterial hypertension	4	3.0	2	2.6	0	0.0	4	7.1	=0.221
Depression	9	6.7	5	6.5	7	15.6	16	28.6	=1.48 <0.001
Mental disease	3	2.2	5	6.5	2	4.4	7	12.7	=1.43 =0.039
Thyroid disease	14	10.4	7	9.1	6	13.3	10	17.9	=0.442
Diabetes	1	0.7	1	1.3	1	2.2	5	8.0	=1.82 =0.010
Allergy	27	20.1	13	16.9	16	35.6	21	37.5	=1.22 =0.008
Acne	4	3.0	4	5.2	2	4.4	3	5.4	=0.830
Autoimmune disease	3	2.2	2	2.6	5	11.1	3	5.4	=0.061
Hormonal disorder	11	8.2	5	6.5	3	6.7	7	12.5	=0.619

* *aOR*—as above. ** Subjects who used an agent versus the subjects who did not.

## Data Availability

Data available on request due to restrictions privacy or ethical. The data presented in this study are available on request from the corresponding author. The data are not publicly available due to restrictions privacy.

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
