# Peer review of "Prevalence of Dry Eye Symptoms and Associated Risk Factors among University Students in Poland"

_ijerph, 2023, doi:10.3390/ijerph20021313_

Round 1

Reviewer 1 Report

This article may represent an interesting addition to the topic of dry eye symptoms, however the limitations of exploring dry eye symptoms versus dry eye disease and other reasons than DED giving rise to these symptoms have to be considered (such as asthenopia), and the reporting on the statistics needs to be expanded.

Please address the following points:

Title not appropriate – the article only investigated dry eye symptoms without confirmation of clinical signs of dry eyes -  without clinical tests, it is not possible to test for prevalence of dry eyes.

 Introduction / Discussion

You say that there is a limited number of studies on the characteristics and prevalence of DED in young people – hence this is the topic of this article, a review of to date published papers regarding this would be appropriate.

Please correct throughout the manuscript ‘DED’ to ‘dry eye symptoms’.

 The study aim is not accurately defined: this study does not investigate the prevalence of DED, however only dry eye symptoms

 Page 6: dry eye syndrome prevalence:

Again, please refrain from DED diagnosis based solely on dry eye questionnaires. The questionnaires only report back dry eye symptoms, however without clinical tests this is not sufficient for DED diagnosis.

Have you considered that asthenopia may also cause some of the symptoms mentioned in the OSDI and DEQ-5 questionnaire? There may be a considerable overlap in reporting. The questionnaire should have at least included some questions exploring asthenopia, in order to exclude that population from the DED group.

>> please discuss this limitation of the study.

Another limitation represents that DED may also be present in participants who did not report any dry eye symptoms – please discuss that you did not account for this subgroup in your study.

Risk factors, medications and comorbidities:

It would be very helpful to add some weight to the individual variables that were found to have a significant impact on OSDI and DEQ-5 score. Can you extract more meaningful information from your multifactor ordinal logistic regression model?

Tables: It is much easier to visualise the descriptive data in form histograms and boxplots. Can you replace some of the tables with graphics?

Were there any other significant correlations next to the obvious one between OSDI and DEQ-5?

Conclusions: ‘the results correspond with a growing pool of evidence that DED symptoms are more common in young adults than previously thought – as mentioned above, please discuss all the relevant studies on dry eye symptoms in young adults and compare their results with yours.

Author Response

Dear Reviewer,

Thank you for giving us the opportunity to submit a revised draft of our manuscript titled ‘The assessment of Dry Eye Disease prevalence and associated risk factors among university students in Poland.’ to International Journal of Environmental Research and Public Health.

We appreciate the time and effort that you have dedicated to providing your valuable feedback on our manuscript. We are grateful to the reviewers for their insightful comments on our paper. We have been able to incorporate changes to reflect most of the suggestions provided by the reviewers. We have highlighted the changes in red within the manuscript.

Here is a point-by-point response to the reviewers’ comments and concerns.

Sincerely,

The authors of paper ijerph-2047275

Author comment: All changes made to the manuscript were marked in red

Reviewers' comment :
This article may represent an interesting addition to the topic of dry eye symptoms, however the limitations of exploring dry eye symptoms versus dry eye disease and other reasons than DED giving rise to these symptoms have to be considered (such as asthenopia), and the reporting on the statistics needs to be expanded.

Please address the following points:

Title not appropriate – the article only investigated dry eye symptoms without confirmation of clinical signs of dry eyes -  without clinical tests, it is not possible to test for prevalence of dry eyes.

Response 1:  We totally agree with the reviewer suggestion. The title of the manuscript has been changed according to the reviewer comment as followed “The assessment of Dry Eye Symptoms prevalence and associated risk factors among university students in Poland.”

Reviewers' comment 2:

 Introduction / Discussion; You say that there is a limited number of studies on the characteristics and prevalence of DED in young people – hence this is the topic of this article, a review of to date published papers regarding this would be appropriate.

Comment:

Before preparing the article we reviewed the literature on the characteristics of dry eye syndrome in young population, especially students. We searched for studies similar to our in the Medline database, using the following keywords „dry eye”, „dry eye disease”, „dry eye syndrome” „dry eye symptoms” and „students”, „college students”, „university students”. We found approximately 30 cohort studies that have been published since 2012, but they differ in many aspects – types of questionnaire, questionnaire cut-off points, analysis methods. Additionally, we would like to point out that most of these studies came from the pre-covid era, so the conditions of the study were different than ours. This makes them difficult to compare. However, when we managed to compare our results with the results of previous research, we discussed them in the „discussion” section. To sum up, based on our search from the present time and current condition, there are not so many publications devoted to this topic, hence our idea this work. The comment was added to the manuscript.

Reviewers' comment 3:

Please correct throughout the manuscript ‘DED’ to ‘dry eye symptoms’.

Response: We thank the Reviewer for catching these confusing errors. Unfortunately, there's a verbal lapse. We really apologise for itThe changes were made as suggested by the Reviewer.

Reviewers' comment 4:

 The study aim is not accurately defined: this study does not investigate the prevalence of DED, however only dry eye symptoms

Response: The aim of the study was to assess the prevalence and risk factors of dry eye symptoms among Polish university students. The corresponding changes were made to the manuscript.

Reviewers' comment 5:

Page 6: dry eye syndrome prevalence: Again, please refrain from DED diagnosis based solely on dry eye questionnaires. The questionnaires only report back dry eye symptoms, however without clinical tests this is not sufficient for DED diagnosis.

 Response: We completely agree with the reviewer on this point. The duplication of this type of errors results from a certain verbal lapse for which we sincerely apologize. Appropriate changes were made to the manuscript.

Our study was based on the online available questionnaire, we did not perform clinical tests, that is limitation of our work. We wanted to show that theoretically healthy young respondents have ailments related to dry eye syndrome. Additionally, to make the concern more distinct we have added a sentence regarding this in the Discussion section:

 “The specific, validated objective clinical tests for dry eye disease assessment were not performed, this is a limitation of the study. Our survey concerned only reported subjective symptoms of dry eye.”.

Reviewers' comment 6:

Have you considered that asthenopia may also cause some of the symptoms mentioned in the OSDI and DEQ-5 questionnaire? There may be a considerable overlap in reporting. The questionnaire should have at least included some questions exploring asthenopia, in order to exclude that population from the DED group.

>> please discuss this limitation of the study.

 Response: We absolutely agree with the reviewer that common visual symptoms in computer vision syndrome (CVS)  is asthenopia. Its symptoms can mimic those associated with dry eye disease, including pain, burning, stinging, irritation, fatigue secondary to extensive computer use, headaches, stiff neck. According to the literature the prevalence of symptoms due to digital eye strain is estimated to range from 25 to 93 per cent, because using of digital devices has become an essential part of everyday life. It is very complex problem with different types as vision-related, oculomotor-related, dry eye or ocular surface related, extraocular or environmental factor-related or decive-related, thus in our opinion it was impossible to detect asthenopia conducting only online survey. Due to this fact we are not even mention about it. We think that the proper diagnosis should  be based on the good standard eye examination, considering the current patients visual demands, habbits, living area ect.

Reviewers' comment 7:

Another limitation represents that DED may also be present in participants who did not report any dry eye symptoms – please discuss that you did not account for this subgroup in your study.

  Response: “We are aware there are some patients with dry eye syndrome without subjective symptoms, due to the fact that we did not perform objective clinical tests, this group of patients may have been omitted in our work, which is another limitation of our study.  However, we would like to point out that many studies show a lack of independence between subjective and objective symptoms, that is why our work is focused on subjective symptoms affecting the comfort and quality of life of our respondents. “  The corresponding comment was posted in the discussion section.

Reviewers' comment 8:

Risk factors, medications and comorbidities: It would be very helpful to add some weight to the individual variables that were found to have a significant impact on OSDI and DEQ-5 score. Can you extract more meaningful information from your multifactor ordinal logistic regression model?

 Response: Multifactor analysis showed that the greatest influence on the higher score both in OSDI and DEQ-5 had stress level (respectively p=0,012 and p=0,009). Mental diseases were proved to be a significant factor in OSDI multifactor analysis (p=0,038), while in DEQ-5 multifactor analysis, other factors showed to be significant were gender (p=0,020) and place of residence (p=0,046).

The comments were added into results section.

Reviewers' comment 9:

Tables: It is much easier to visualise the descriptive data in form histograms and boxplots. Can you replace some of the tables with graphics?

 Response:   Due to the large amount of data, we decided to present them mainly in tables so that the results are collected in one place and seem to be easier for interpreting. We added a histogram about stress level among the responders.

Reviewers' comment 10:

Were there any other significant correlations next to the obvious one between OSDI and DEQ-5?

 Response: By showing a strong relationship between the answers obtained in both questionnaires, we wanted to show that the respondents gave reliable, honest answers, because they are similar, not random, and therefore it can be concluded that we conducted a reliable study. We added some other significant correlations in the result section.

Reviewers' comment 11:

Conclusions: ‘the results correspond with a growing pool of evidence that DED symptoms are more common in young adults than previously thought – as mentioned above, please discuss all the relevant studies on dry eye symptoms in young adults and compare their results with yours.

Response: As mentioned above – during the literature review, we found approximately 30 similar cohort studies which have been published since 2012. Most of them were conducted before the COVID-19 pandemic, thereby before the popularisation of remote learning and extending the time of using digital display terminals. We believe that this is still not enough data about the epidemiology of dry eye syndrome in students. Furthermore, in the most comprehensive report on dry eye disease  „TFOS DEWS II Epidemiology Report”  (in our article – reference no. 2) the limited data and great need for further research on dry eye in the population younger than 40 years old was highlighted. After the publication of this report, the number of studies regarding students increased. We mentioned them in our article’s discussion to compare their results with ours.

Reviewer 2 Report

The study lacks novelty. Here are some queries.

Introduction:

Line 32-34 – the logical flow of this paragraph should be improved, it seems inappropriate to place the two sentences about prevalence in the middle of ocular symptoms.

Line 45 –digital display terminals (VDT):Should it be abbreviated as DDT?

The novelty of the study should be better explained, the authors indicate that there is a lack of research about the characteristics of DED in younger populations, but from the discussion it seems that there are more similar studies in student populations.

Materials and Methods:

Were the questionnaires used in the study authorized by the authors? (OSDI, DEQ-5)

The sample size calculation should be supplemented.

Questionnaire content should be explained in more detail. For example, what is the specific options for presence of systemic diseases and taken medications.

Why did the authors use two questionnaires to measure DED, is there any advantage over using one questionnaire? Using two questionnaires at the same time makes the interpretation of the results more cumbersome, which seems unnecessary, after all, the purpose of this study is not to compare the two questionnaires.

Before performing ordinal logistic regression, a parallel line test should be done.

Results:

Table 1 – the definition of smoking habit is not clear.

Page 5 – what is the 1-5 stress level scale? Neither in the methods section nor in the table is there a description of the stress level.

The results of these section (Socio-demographic and clinical risk factors, Medications and comorbidities) should be shown in the table.

All tables should be better adjusted like using three-line tables.

Figure captions are generally placed below the figures.

Discussion

Page 11Line 48-51 – Were these risk factors (refractive errors, contact lenses) included in the regression analysis at the same time? Were covariates (such as socio-demographic factors) added to the regression model?

Page 13 – Line 7-9 – This sentence gets a little out of place in this part of the discussion, when the context is all about how stress level affects DED.

Page 13 – Line 13-39 – Although the relationship between DED and mental disease may be bidirectional, too much discussion in a cross-sectional study may be unnecessary.

Author Response

Dear Reviewer,

Thank you for giving us the opportunity to submit a revised draft of our manuscript titled ‘The assessment of Dry Eye Disease prevalence and associated risk factors among university students in Poland.’ to International Journal of Environmental Research and Public Health.

We appreciate the time and effort that you have dedicated to providing your valuable feedback on our manuscript. We are grateful to the reviewers for their insightful comments on our paper. We have been able to incorporate changes to reflect most of the suggestions provided by the reviewers. We have highlighted the changes in red within the manuscript.

Here is a point-by-point response to the reviewers’ comments and concerns.

Sincerely,

The authors of paper ijerph-2047275

Author comment: All changes made to the manuscript were marked in green

Reviewer Comment 1:

Introduction:  Line 32-34 – the logical flow of this paragraph should be improved, it seems inappropriate to place the two sentences about prevalence in the middle of ocular symptoms.

Response : We thank the Reviewer for catching this inconvenience.  The corresponding changes have been made to the manuscript.

Comment 2 : Line 45 –digital display terminals (VDT)Should it be abbreviated as DDT?

Response: The change was made as suggested by the Reviewer.

Comment 3 :

The novelty of the study should be better explained, the authors indicate that there is a lack of research about the characteristics of DED in younger populations, but from the discussion it seems that there are more similar studies in student populations.

 Response:

Before preparing the article we reviewed the literature on the characteristics of dry eye syndrome in young population, especially students. We searched for studies similar to our in the Medline database, using the following keywords „dry eye”, „dry eye disease”, „dry eye syndrome” „dry eye symptoms” and „students”, „college students”, „university students”. We found approximately 30 cohort studies that have been published since 2012, but they differ in many aspects – types of questionnaire, questionnaire cut-off points, analysis methods. Additionally, we would like to point out that most of these studies came from the pre-covid era , thereby before the popularisation of remote learning and extending the time of using digital display terminals, so the conditions of the study were different than ours. This makes them difficult to compare. We believe that this is still not enough data about the epidemiology of dry eye syndrome in students. Furthermore, in the most comprehensive report on dry eye disease  „TFOS DEWS II Epidemiology Report”  (in our article – reference no. 2) the limited data and great need for further research on dry eye in the population younger than 40 years old was highlighted. After the publication of this report, the number of studies regarding students increased. We mentioned them in our article’s discussion to compare their results with ours.  

Comment 4 :

Materials and Methods: Were the questionnaires used in the study authorized by the authors? (OSDI, DEQ-5)

Response: In our opinion it was not necessary to authorized both mentioned questionnaires because they were validated and are widely available in Polish version.

Comment 5 : Materials and Methods: The sample size calculation should be supplemented.

Response: It is estimated that the total number of university students in Poland in 2022 accounted to 1,2 million, including 40 thousand students of faculties of medicine. According to relevant data, the estimated prevalence of DED in adult population in Poland was 10%-18% and up to 30% in certain groups, i.e., mainly in students. Hence, the sampling frame was 30% out of 40000, i.e., 12000 individuals. Assuming the margin of error at 5% and 95% confidence level, the required sample size of the present survey accounted at least 321 subjects. Despite the relatively long time of, and our effort put into, the data collection process, the Investigators managed to obtain complete questionnaires from 312 respondents, which constituted 97,2% of the required sample size we previously had computed. The difference did not exceed the 5% threshold, which is why the Authors decided to conduct statistical analyses of the answers from 312 respondents.

Comment 6 : Materials and Methods: Questionnaire content should be explained in more detail. For example, what is the specific options for presence of systemic diseases and taken medications.

Response:  In the conducted survey, apart from the from the OSDI and DEQ-5 questionnaire, we included our own authorship questions, that aimed to collect  socio-demographic data (gender, age group, place of residence, field of study and year of study), behavioral and environmental factors (use of artificial tears, lifestyle with special emphasis on using electronic devices, spending time outdoors and in air-conditioned interiors), risk factors of dry eye syndrome (refractive error, previous refractive surgery procedures, use and tolerance of contact lenses, history of eye injuries, presence of eye diseases, presence of systemic diseases and taken medications, addictions, history of SARS-CoV-2 infection, vaccination against COVID-19 and stress level).  The questions concerning presence of systemic diseases and taken medications looked like below:

Do you have the following diseases?

  1. a) hypertension
  2. b) heart diseases
  3. c) depression
  4. d) other mental illnesses
  5. e) thyroid disease
  6. f) diabetes
  7. g) allergy
  8. h) rosacea
  9. i) autoimmune diseases (RA, lupus, systemic sclerosis, AS, Sjogren's syndrome, others)
  10. j) vitamin A deficiency
  11. k) neurological diseases (MS, neuropathies)
  12. l) amyloidosis
  13. m) sarcoidosis
  14. n) hormonal disorders

Which of the following drugs are you taking on a regular basis?

  1. a) β-blockers
  2. b) diuretics
  3. c) psychotrop medicines
  4. d) antihistamines
  5. e) painkillers
  6. f) glucocorticoids (GCS)
  7. g) anabolic steroids
  8. h) oral contraceptives
  9. i) other hormonal drugs

We can add the file  with the questions as supplementary material to the article.

 Comment 7:

Why did the authors use two questionnaires to measure DED, is there any advantage over using one questionnaire? Using two questionnaires at the same time makes the interpretation of the results more cumbersome, which seems unnecessary, after all, the purpose of this study is not to compare the two questionnaires.

Response:

According to the literature both of the questionnaires have high sensitivity and specificity in discriminating symptoms of dry eye thus are valid assessment tools in both clinical and epidemiological studies. [ Akowuah PK, Adjei-Anang J, Nkansah EK, Fummey J, Osei-Poku K, Boadi P, Frimpong AA. Comparison of the performance of the dry eye questionnaire (DEQ-5) to the ocular surface disease index in a non-clinical population. Cont Lens Anterior Eye. 2022 Jun;45(3):101441. doi: 10.1016/j.clae.2021.101441. Epub 2021 Apr 6. PMID: 33836971., Pult H, Wolffsohn JS. The development and evaluation of the new Ocular Surface Disease Index-6. Ocul Surf. 2019 Oct;17(4):817-821. doi: 10.1016/j.jtos.2019.08.008. Epub 2019 Aug 20. PMID: 31442595.]

As the reviewer observed, our goal was not to compare these questionnaires. Considering that we only conducted a survey without a clinical examination of the respondents, we used both questionnaires to check the reliability and credibility of the answers given by the respondents, taking into account the sensitivity and specificity of these tests.

Comment 8:

Before performing ordinal logistic regression, a parallel line test should be done.

Response: We are aware that we should have performed this particular test. Unfortunately, we did not have position to carry out this procedure. The reason is that  the statistic package we used on regular basis at  the university does not contain this particular routine. Therefore, the obstacle is objective.

 Comment 9:

Results: Table 1 – the definition of smoking habit is not clear.

Response: We have replaced the smoking habit with a regular, daily- basis use tobacco products, as it was posted in the survey question, to make it easier for the reader to interpret.

Comment  10:

Page 5 – what is the 1-5 stress level scale? Neither in the methods section nor in the table is there a description of the stress level.

Response: Thank you very much for your valuable suggestion. The relevant information was introduced into the manuscript. The question about stress looks like below”

How would you rate your overall stress level?

0 Lack of stress

1

2

3

4

5 Maximum stress,

Thus, it is was self-assessment question about stress level between 0-lack of stress and 5 -maximum of stress.

Comment 11:

The results of these section (Socio-demographic and clinical risk factors, Medications and comorbidities) should be shown in the table.

Response: We thank the Reviewer for pointing this out. Additional details and tables were added to the manuscript according to the Reviewer suggestion.

Comment 12 :

All tables should be better adjusted like using three-line tables.

Response: We have changed the format of the tables according to the Reviewer suggestion.

Comment 13:

Figure captions are generally placed below the figures.

Response:  We appreciate the Reviewer suggestion. Some details have been added to the figure legends in the manuscript and we hope that it is now more clear.

Comment 14:

Discussion, Page 11 – Line 48-51 – Were these risk factors (refractive errors, contact lenses) included in the regression analysis at the same time? Were covariates (such as socio-demographic factors) added to the regression model?

Response: Yes, there were carried out multivariate regression models which included the aforementioned risk factors and controlled for the patients’ socio-demographic traits.

Comment 15:

Page 13 – Line 7-9 – This sentence gets a little out of place in this part of the discussion, when the context is all about how stress level affects DED.

Response: We thank the reviewer for pointing this out. We have decided to deleted the sentence: “ On the other hand, patients with DED are more prone to developing severe psychological stress [7], as constant feeling of discomfort and distress can lead to mood disorders [46].  

Comment 16 :

Page 13 – Line 13-39 – Although the relationship between DED and mental disease may be bidirectional, too much discussion in a cross-sectional study may be unnecessary.

Response:

We took this suggestion into account and reorganized the section devoted to the topic.

Round 2

Reviewer 1 Report

Many thanks for the many improvements to your paper.

I feel however, that not all comments were adequately addressed.

Title: please consider changing it to the following wording:

Prevalence of dry eye symptoms and associated risk factors among university students in Poland.

Please reword the new section in Methods and section 4.5 by avoiding the use of the first person. In fact, the entire article requires scanning for first person use.

What is a simple size calculation (section 2.1) and how does this apply to a literature search?

Please consult a statistician for extraction of some weighting of the individual variables that were found to have a significant impact on OSDI And DEQ-5 scores from the multifactorial ordinal logistic regression model. P-values only give an indication to whether a factor is statistically significant, however not how one variable affects the outcome variable, compared to another.

Author Response

Dear Reviewer,

Thank you for giving us the opportunity to submit a revised draft of our manuscript titled ‘The assessment of Dry Eye Disease prevalence and associated risk factors among university students in Poland.’ to International Journal of Environmental Research and Public Health.

We appreciate the time and effort that you have dedicated to providing your valuable feedback on our manuscript. We are grateful to the reviewers for their insightful comments on our paper. We have been able to incorporate changes to reflect most of the suggestions provided by the reviewers. We have highlighted the changes in red within the manuscript.

Here is a point-by-point response to the reviewers’ comments and concerns.

Sincerely,

The authors of paper ijerph-2047275

Reviewer 1

All changes made to the manuscript were marked in purple

Comment 1

Title: please consider changing it to the following wording:

Prevalence of dry eye symptoms and associated risk factors among university students in Poland.

Response:  The title has been changed according to the Reviewer suggestion as below: Prevalence of dry eye symptoms and associated risk factors among university students in Poland.

Comment 2

Please reword the new section in Methods and section 4.5 by avoiding the use of the first person. In fact, the entire article requires scanning for first person use.

Response:  The changes were made according to the Reviewer suggestion.

Comment 3

What is a simple size calculation (section 2.1) and how does this apply to a literature search?

Response:  The comment was added to the manuscript in Materials and Methods, 2.1 Simple size calculation as below:

It is estimated that the total number of university students in Poland in 2022 accounted to 1,2 million, including 40 thousand students of faculties of medicine.  The mentioned data comes from the report of the Central Statistical Office in Poland. (Central Statistical Office is a central government administration office dealing with the collection and sharing of statistical information on most areas of public life and some aspects of private life.) According to relevant data, the estimated prevalence of DED in adult population in Poland was 10%-18% and up to 30% in certain groups, i.e., mainly in students. Hence, the sampling frame was 30% out of 40000, i.e., 12000 individuals. Assuming the margin of error at 5% and 95% confidence level, the required sample size of the present survey accounted at least 321 subjects. Despite the relatively long time of, and our effort put into, the data collection process, the Investigators managed to obtain complete questionnaires from 312 respondents, which constituted 97,2% of the required sample size we previously had computed. The difference did not exceed the 5% threshold, which is why the Authors decided to conduct statistical analyses of the answers from 312 respondents.

Comment 4

Please consult a statistician for extraction of some weighting of the individual variables that were found to have a significant impact on OSDI And DEQ-5 scores from the multifactorial ordinal logistic regression model. P-values only give an indication to whether a factor is statistically significant, however not how one variable affects the outcome variable, compared to another.

Response:  We added the effect size in the form of odds ratio (OD) in the tables. This shows how strong are the presented dependencies.

Reviewer 2 Report

1.The content of the sample size calculation was not added to the main text.

2.Many other statistical software can complete the parallel line test, such as stata, R, etc.

3.Line206-207- Table 2 and 3 should add more columns containing DEQ-5 scores and P values (line 203 - 204).

4.Variables adjusted in the regression model can be annotated in the corresponding tables.

Author Response

Dear Reviewer,

Thank you for giving us the opportunity to submit a revised draft of our manuscript titled ‘The assessment of Dry Eye Disease prevalence and associated risk factors among university students in Poland.’ to International Journal of Environmental Research and Public Health.

We appreciate the time and effort that you have dedicated to providing your valuable feedback on our manuscript. We are grateful to the reviewers for their insightful comments on our paper. We have been able to incorporate changes to reflect most of the suggestions provided by the reviewers. We have highlighted the changes in red within the manuscript.

Here is a point-by-point response to the reviewers’ comments and concerns.

Sincerely,

The authors of paper ijerph-2047275

Reviewer 2

All changes made to the manuscript were marked in blue

Comment 1

The content of the sample size calculation was not added to the main text.

Response:   The comment was added to the manuscript in Materials and Methods, 2.1 Simple size calculation as below: It is estimated that the total number of university students in Poland in 2022 accounted to 1,2 million, including 40 thousand students of faculties of medicine.  The mentioned data comes from the report of the Central Statistical Office in Poland. (Central Statistical Office is a central government administration office dealing with the collection and sharing of statistical information on most areas of public life and some aspects of private life.) According to relevant data, the estimated prevalence of DED in adult population in Poland was 10%-18% and up to 30% in certain groups, i.e., mainly in students. Hence, the sampling frame was 30% out of 40000, i.e., 12000 individuals. Assuming the margin of error at 5% and 95% confidence level, the required sample size of the present survey accounted at least 321 subjects. Despite the relatively long time of, and our effort put into, the data collection process, the Investigators managed to obtain complete questionnaires from 312 respondents, which constituted 97,2% of the required sample size we previously had computed. The difference did not exceed the 5% threshold, which is why the Authors decided to conduct statistical analyses of the answers from 312 respondents.

Comment 2

Many other statistical software can complete the parallel line test, such as stata, R, etc.

Response:  We are aware about other statistical software can complete the parallel line test, such as stata, R, etc. Unfortunately, we are employees of a public university that only has a license for the Statistica program, thus we are unable to formally use other software. Moreover, we believe that the study group is so large that it does not require the use of other tests.

Comment 3

Line206-207- Table 2 and 3 should add more columns containing DEQ-5 scores and P values (line 203 - 204).

Response:  The tables were adjusted with columns containing DEQ-5 and OSDI scores with p values.

Pharmaceutical agent

Frequency

DEQ-5 score

OSDI score

n

%

M

SD

p-value

M

SD

p-value

Beta-blockers

6

1,9

11,2

6,6

= 0,159

31,6

35,4

= 0,482

Psychotropics

38

12,2

9,4

5,2

= 0,014

30,1

23,1

= 0,002

Anti-histamines

33

10,6

9,4

5,7

= 0,054

28,5

23,3

= 0,024

Analgesics

18

5,8

7,7

5,8

= 0,923

30,4

26,7

= 0,062

Glucocorticosteroids

5

1,6

13

5,2

= 0,026

44,8

33,4

= 0,024

Contraceptives

47

15,1

9,1

5,3

= 0,023

24,7

20,6

= 0,127

Other hormones

34

10,9

8,8

5,4

= 0,170

26,7

23,7

= 0,152

Tab. 2. Medication taken chronically by the study participants (n = 312 *)

Health condition

Frequency

DEQ-5 score

OSDI score

n

%

M

SD

p-value

M

SD

p-value

Arterial hypertension

10

3,2

10,1

6,2

= 0,165

29,6

29,0

= 0,299

Heart disease

6

1,9

9,0

7,5

= 0,778

30,7

35,7

= 0,620

Depression

37

11,9

9,7

5,2

= 0,005

30,6

20,8

< 0,001

Mental disease

17

5,4

9,6

5,9

= 0,127

34,0

28,0

= 0,019

Thyroid disease

37

11,9

8,5

5,4

= 0,219

26,8

24,8

= 0,155

Diabetes

8

2,6

13,9

5,1

= 0,001

44,8

29,7

= 0,008

Allergy

77

24,7

8,7

4,7

= 0,023

25,4

19,6

= 0,002

Acne

13

4,2

9,5

6,2

= 0,242

30,3

29,0

= 0,210

Tab. 3. Comorbidities in the study participants (n = 312 *)

Comment 4

Variables adjusted in the regression model can be annotated in the corresponding tables.

Response:  Considering the multivariate regression model performed we would like to underline that all the dependent variables were adjusted. The corresponding changes have been made to the tables.

Independent trait

Normal

Mild

Moderate

Severe

aOR*

p-value

n

%

n

%

n

%

n

%

How many hours a day do you spend in close proximity to the screen of an electronic device, e.g. mobile phone, computer, tablet, reader (not taking into account a TV set)?:

< 4

29

21,6

11

14,3

6

13,3

3

5,4

= 1,37

= 0,006

4-8

65

48,5

49

63,6

24

53,4

25

44,6

> 8

40

29,9

17

22,1

15

33,3

28

50,0

How many hours a day do you spend in front of the screen of an electronic device (not taking into account a TV set), without taking breaks?

< 1

32

23,9

6

7,8

8

17,8

5

8,9

= 1,16

= 0,016

1-2

56

41,8

43

55,8

19

42,2

23

41,1

3-4

26

19,4

23

29,9

11

24,4

19

33,9

> 4

20

14,9

5

6,5

7

5,6

9

16,1

How many hours a day do you spend outdoors?

< 1

39

29,1

25

32,5

16

35,6

21

37,5

= 0,261

1-4

75

56,0

48

62,3

22

49,9

31

55,4

> 4

20

14,9

4

5,2

7

15,6

4

7,1

How many hours a day do you spend in air-conditioned interiors?

< 1

83

61,9

44

57,1

24

53,3

34

60,8

= 0,648

1-4

30

22,4

21

27,3

9

20,0

11

19,6

> 4

21

15,7

12

15,6

12

26,7

11

19,6

regular, daily basis use tobacco products

32

23,9

11

14,3

15

33,3

16

28,6

= 0,078

History of SARS-CoV-2 infection

75

56,0

34

44,2

25

55,6

39

69,6

= 0,036

aOR – adjusted odds ratio; the adjutancy procedure comprised the participants’ age, gender, place of residence, faculty, refraction error, use of contact lenses, comorbidities, medicines taken.)

Tab. 4. Severity of dry eye, according to the OSDI, in the study participants by selected behavioural and environmental factors (n = 312)

Pharmaceutical agent

Normal

Mild

Moderate

Severe

aOR*

p-value**

n

%

n

%

n

%

n

%

Psychotropics

9

6,7

8

10,4

6

13,33

15

26,8

= 1,41

= 0,002

Anti-histamines

12

9,0

5

6,5

6

13,3

10

17,9

= 0,156

Analgesics

6

4,5

3

3,9

4

8,9

5

8,9

= 0,433

Contraceptives

(females only)

19

23,7

8

12,5

7

20,6

12

29,3

= 0,184

Other hormones

13

9,7

7

9,1

5

11,1

9

16,1

= 0,572

(* aOR – as above. ** Subjects who used an agent versus the subjects who did not.)

Tab. 5. Severity of dry eye, according to the OSDI, in the study participants by medication (n = 312)

Health condition

Normal

Mild

Moderate

Severe

aOR*

p-value**

n

%

n

%

n

%

n

%

Arterial hypertension

4

3,0

2

2,6

0

0,0

4

7,1

= 0,221

Depression

9

6,7

5

6,5

7

15,6

16

28,6

= 1,48

< 0,001

Mental disease

3

2,2

5

6,5

2

4,4

7

12,7

= 1,43

= 0,039

Thyroid disease

14

10,4

7

9,1

6

13,3

10

17,9

= 0,442

Diabetes

1

0,7

1

1,3

1

2,2

5

8,0

= 1,82

= 0,010

Allergy

27

20,1

13

16,9

16

35,6

21

37,5

= 1,22

= 0,008

Acne

4

3,0

4

5,2

2

4,4

3

5,4

= 0,830

Autoimmune disease

3

2,2

2

2,6

5

11,1

3

5,4

= 0,061

Hormonal disorder

11

8,2

5

6,5

3

6,7

7

12,5

= 0,619

(* aOR – as above. ** Subjects who used an agent versus the subjects who did not.)

Tab. 6. Severity of dry eye, according the OSDI, in the study participants by comorbidity (n = 312)
